# TOWARDS MAKING LINEAR ATTENTION USABLE

## ABSTRACT

The original softmax based "all-to-all" attention mechanism in the extremely successful transformer architecture computes the attention between $N$ tokens embedded in a $D$-dimensional head, in $O(N^2D)$ time and $O(N^2)$ memory. Several linearly scaling attention mechanisms have been proposed over the seven years since the transformer algorithm was proposed, to address the quadratic complexity in $N$, and these "Linear" algorithms have been shown to achieve reasonable training performance in the papers they were introduced in. Despite the fact that most current LLM applications are trending towards processing larger token sequences in context (increasing $N$ into the millions and beyond), and transformers have helped create foundational models in diverse domains beyond language (e.g., image, video, and audio processing), the proposed linear mechanisms surprisingly do not find wide usage, and moreover remain restricted to relatively small sized problems. We address two possible reasons for this. The first is that many of the proposed mechanisms, despite their linear complexity in $N$, have practically a relatively large memory footprint which is quadratic in $D$, which quickly fills memory of available GPUs. We show the memory complexity can be reduced by approaching calculations from a novel perspective in both the forward and backward passes in training. A second reason is that techniques that accelerate convergence, such as dropout, cannot be used during training in the current linear algorithms. This is mitigated via an alternative dropout mechanism. Results and comparison studies demonstrate the usefulness of the proposed approaches, while maintaining the linear scaling in $N$ in both wall-clock time and memory usage.

## 1 INTRODUCTION

Transformers are the single deep learning architecture that underpin many recent successful applications in diverse fields such as natural language processing, speech, computer vision, biology, and so on. Transformers incorporate a Softmax-based all-to-all "Attention" mechanism between the input tokens Vaswani et al. (2017). While this has proven to be extraordinarily effective in learning tasks, it has a time and memory complexity of $O(N^2D)$ and $O(N^2)$ respectively, where $N$ is the number of tokens and $D$ the dimension per attention head. Efficient hardware implementation can offer constant speedup, and significant memory reduction. A notable example is the widely used FlashAttention-2 Dao (2023), where the memory complexity is reduced to $O(ND)$, but the time complexity remains $O(N^2D)$. Given how the industrial models are shifting towards ever larger $N$ ($\sim 10^7$ in current implementations), searching for a linearly scaling attention mechanism started shortly after the introduction of the original paper.

Previous proposals for a linearly scaling attention mechanism used either *Sparsified/localized Attention*, or *Kernel Separation*. In the former, the all-to-all attention, and as a result long-range context is not captured. In the former approach, all-to-all attention is not captured, resulting in a loss of long-range context. While this may be effective in certain situations (particularly for approximations during inference), it compromises the model's ability to learn from larger sequence, which was the goal in the first place. Kernel Separation, in contrast, successfully captures all-to-all attention and has demonstrated promising results on small benchmarks. However, its major drawback is the high memory consumption of $O(ND^2)$. This drawback has hindered its application in large language models (LLMs), limiting researchers' ability to study its effectiveness at scale. To put things in perspective, with a memory consumption of $O(ND^2)$, TinyLlama-1.1B Zhang et al. (2024b) would

require 280GB[1] of memory per batch. *This makes the Kernel Separation approach impractical on any production GPU unless the memory consumption is addressed.*

We propose a new viewpoint for calculating Linear Attention, and show how the memory consumption can be reduced to $O(ND)$. Using our method, the memory consumption of TinyLlama-1.1B with Linear Attention is reduced to 2.2 GB memory. **Our work does not introduce a new Attention mechanism, and our focus is addressing the high memory consumption issue in the many linear attention mechanisms proposed.** Furthermore, since these methods do not explicitly calculate the attention matrix, dropout cannot be applied. This is problematic given the robustness and accelerated convergence that dropout offers. To address this, we introduce an alternative mechanism. In summary, we make the following contributions:

- Reduce the memory consumption of Linear attention from $O(ND^2)$ to $O(ND)$ (Section 3).
- Introduce an alternative mechanism to dropout (Section 4).

## 2 BACKGROUND AND RELATED WORK

**Notation:** Bolded upper case letters, e.g., $\mathbf{X}$ indicate matrices, and bolded lower case letters $\mathbf{x}_i$ and $\mathbf{x}_{i,j}$ the $i$-th row and the element at the $i$-th row and $j$-th column of $\mathbf{X}$ respectively. $\mathbf{x}_j^T$ denotes the $j$-th row of $\mathbf{X}$. Unbolded letters $X, x, x_i, x_{jkl}^{(i)}$ indicate scalars.

### 2.1 PRELIMINARIES: COMPUTING "VANILLA" ATTENTION

In vanilla attention, given a sequence of $N$ tokens, model dimension of $C$, $H$ heads, and dimension per head of $D = C/H$, each head takes in three matrices $\mathbf{Q}, \mathbf{K}, \mathbf{V} \in \mathbb{R}^{N \times D}$, and gives an output matrix $\mathbf{O} \in \mathbb{R}^{N \times D}$. The output is calculated using a matrix vector product (MVP) of the Attention $\mathbf{A}$ with $\mathbf{V}$ as follows:

$$\mathbf{O} = \mathbf{A}\mathbf{V}, \ \mathbf{A} = \text{Softmax}\left(\mathbf{Q}\mathbf{K}^T\right), \ \mathbf{o_{i,j}} = \frac{\sum_{n=1}^{N} \exp(\mathbf{q}_i.\mathbf{k_n}/\sqrt{D})\mathbf{v_{n,j}}}{\sum_{n=1}^{N} \exp(\mathbf{q}_i.\mathbf{k}_n/\sqrt{D})} = \frac{\sum_{n=1}^{N} f(\mathbf{q}_i.\mathbf{k_n})\mathbf{v_{n,j}}}{\sum_{n=1}^{N} f(\mathbf{q}_i.\mathbf{k}_n)} \quad (1)$$

where $f(x)$ is the attention kernel. Therefore, each head has a computational complexity of $O(N^2D)$. The final output is given by concatenating outputs from each head, with a total $O(N^2C)$ computation. We can apply a causal mask by changing Eq. 1 to

$$\mathbf{O} = \text{tril}(\mathbf{A})\mathbf{V}, \quad \text{tril}(\mathbf{A})_{i,j} = \begin{cases} \mathbf{a}_{i,j}, j \leq i \\ 0, \ j > i \end{cases}, \quad \mathbf{o}_{i,j} = \frac{\sum_{n=1}^{i} f(\mathbf{q}_i.\mathbf{k}_n)\mathbf{v}_{n,j}}{\sum_{n=1}^{i} f(\mathbf{q}_i.\mathbf{k}_n)}. \quad (2)$$

The softmax attention kernel of $f(x) = \exp(q \cdot k/\sqrt{D})$ could be changed. Any definite positive convex function can substitute for the attention kernel, and several functions have been explored in the literature.

### 2.2 RELATED WORK

The most successful approach to speedup the vanilla attention mechanism that has seen widespread use is efficient hardware implementation and the use of analytical gradients. The most notable such mechanism, which offers a constant speedup, and reduces memory consumption to $O(ND)$ is FlashAttention-2 Dao (2023), Dao (2024).

However, this method does not address the issue of quadratic cost of large context attention. Because of this there is a lot of research on systems approaches to use multi-GPU systems, e.g., Liu et al. (2023).

The goal of this paper is to make sub-quadratic attention usable. Two common algorithm-based approaches proposed in the literature for this are Kernel Separation, and Sparsification.

---

[1]Context length $N = 2048$ tokens, head dimension $D = 128$, number of heads $H = 16$, number of levels $l = 22$: Memory= $N \times H \times D^2 \times l \times 2 \times 3 \times 4$. The $\times 2$ comes from having to store kernel coefficients for the Query and Key transformations, the $\times 3$ from the automatic differentiation libraries overhead, and the $\times 4$ is the Float-32 size in Bytes.

**Kernel Separation**  In this approach the attention kernel is chosen to be one that allows $f(\mathbf{q}_i.\mathbf{k}_n)$ to be written as $g(\mathbf{q}_i)^T h(\mathbf{k_n})$, where $g(x)$ and $h(x)$ are deliberately chosen $\mathbb{R}^D \to \mathbb{R}^{D'}$ functions. The output can then be calculated as

$$\mathbf{o_{i,j}} = \frac{g(\mathbf{q}_i)^T \sum_{n=1}^N h(\mathbf{k_n})\mathbf{v_{n,j}}}{g(\mathbf{q}_i)^T \sum_{n=1}^N h(\mathbf{k_n})}, \tag{3}$$

which brings down the calculation complexity to $O(ND)$. Common choices of $f(x)$ are $a + bx$ (e.g., Katharopoulos et al. (2020); Kasai et al. (2021); Zhang et al. (2024a); Choromanski et al. (2020); Qin et al. (2022b)), $a + bx + cx^2$ Keles et al. (2023); Banerjee et al. (2020), with the coefficients either as the Taylor expansion of the exponential or as learnable parameters. The most widely adopted attention kernel is $f(x) = a + bx$, which is also known as Linear Attention. As previously mentioned, Linear Attention has a computational complexity of $O(ND)$ and a memory complexity of $O(ND^2)$.

**Sparsification**  In this approach, we only calculate the Attention for a fixed number of keys for each query. Common approaches consider keys with tokens that are spatially close-by to the query's token Zaheer et al.; Zhou et al.; Yu et al. (2023a); Beltagy et al. (2020); Han et al. (2023), randomized sampling Wiegreffe & Pinter (2019), or taking advantage of locality aware hashing Kitaev et al., where the query and key vectors are hashed based based on their position in the $D-$dimensional space, and attention is only calculated for the keys close to the query with this metric. However, *these approaches contradict the goal of capturing attention for a long sequence, as it only considers limited token pairs.* This is especially problematic for training.

There are works which that combine both of these approaches, where they calculate the exponential-based kernel for a sparse number of query-key pairs, and separable-based kernel for the rest of the tokens Qin et al. (2022a).

**Other Algorithmic Approaches**  Other approaches include assuming that the Attention matrix is numerically low rank and derive a smaller matrix Wang et al., hierarchically packing the tokens Ma et al. (2021), using the Gaussian function as the Attention kernel and approximating it in linear time Chen et al. (2021) or calculating the result of the Attention and the value matrix multiplication using Fast Multipole Methods Nguyen et al. (2021). Despite their novel approaches, these methods have yet to be used in large context LLMs.

**Non-transformer Models**  New learning architectures are also being proposed as an alternative to Transformers. Recent ones which have generated considerable interest include Mamba Gu & Dao, Retentive networks Sun et al., Hyena Poli et al. and CRATE Yu et al. (2023b). These approaches are complementary to the factored attention kernels which are the focus of this paper.

## 3  MEMORY REDUCTION

We first discuss factorization of the attention kernel, which is the computational trick enabling speedup. This enables the time and memory complexity of factorization based kernels can be reduced both in the forward and backward passes. The derivation of the analytical gradient of the attention head in this paper is a novel contribution, which allows us to reduce memory consumption. We show these results for the linear attention kernel $f(x) = a + bx$, and demonstrate how to achieve time and memory complexities of $O(ND^2)$ and $O(ND)$. Our method can also be applied to the attention kernel of $f(x) = a + bx + cx^2$, which we show in Appendix A, and results in $O(ND^3)$ time and $O(ND)$ memory complexity.

### 3.1  SPEEDUP VIA FACTORIZATION

To show how Factorization speeds up computations, consider the MVP

$$\begin{bmatrix} a_1b_1 & a_1b_2 & a_1b_3 \\ a_2b_1 & a_2b_2 & a_2b_3 \\ a_3b_1 & a_3b_2 & a_3b_3 \end{bmatrix} \times \begin{bmatrix} d_1 \\ d_2 \\ d_3 \end{bmatrix} = \begin{bmatrix} u_1 \\ u_2 \\ u_3 \end{bmatrix}. \tag{4}$$

The naive method of calculating $u$ would be

$$u_i = a_ib_1d_1 + a_ib_2d_2 + a_ib_3d_3, \tag{5}$$

with a total of 9 multiplications and 6 accumulations. Applying Factorization, we have

$$x = b_1 d_1 + b_2 d_2 + b_3 d_3, \quad u_i = a_i x, \tag{6}$$

reducing the operations to 6 multiplications and 2 accumulations. For an $M \times M$ with the same structure, the number of operations reduces from $O(M^2)$ to $O(M)$.

## 3.2 FORWARD PASS

Employing $f(x) = a + bx$ as Attention kernel, the output matrix $\mathbf{O}$ can be broken down to matrix multiplications in the form of Eq. 4. Specifically,

$$\mathbf{o}_{i,j} = \frac{\sum_{n=1}^{N} f(\mathbf{q}_i^T \mathbf{k_n}) \mathbf{v}_{n,j}}{\sum_{n=1}^{N} f(\mathbf{q}_i^T \mathbf{k_n})}, \quad \mathbf{o}_{i,j} = \frac{\mathbf{f}_{i,j}}{\mathbf{g}_i}, \quad \mathbf{F} \in \mathbb{R}^{N \times D}, \mathbf{G} \in \mathbb{R}^N, \tag{7}$$

where $\mathbf{f}_{i,j}$ and $\mathbf{g}_i$ are elements of $\mathbf{F}$ and $\mathbf{G}$. Specifically, $\mathbf{F}$ and $\mathbf{G}$ are written as

$$\mathbf{F} = \left( a + b \sum_{m=1}^{D} \begin{bmatrix} \mathbf{q}_{1,m}\mathbf{k}_{1,m} \cdots \mathbf{q}_{1,m}\mathbf{k}_{N,m} \\ \vdots \ddots \vdots \\ \mathbf{q}_{N,m}\mathbf{k}_{1,m} \cdots \mathbf{q}_{N,m}\mathbf{k}_{N,m} \end{bmatrix} \right) \mathbf{V} \tag{8}$$

$$\mathbf{G} = \left( a + b \sum_{m=1}^{D} \begin{bmatrix} \mathbf{q}_{1,m}\mathbf{k}_{1,m} \cdots \mathbf{q}_{1,m}\mathbf{k}_{N,m} \\ \vdots \ddots \vdots \\ \mathbf{q}_{N,m}\mathbf{k}_{1,m} \cdots \mathbf{q}_{N,m}\mathbf{k}_{N,m} \end{bmatrix} \right) \mathbb{1} \tag{9}$$

where $\mathbb{1} \in \mathbb{R}^N$ is a vector of all ones; i.e.,

$$\mathbf{f}_{i,j} = \sum_{n=1}^{N} \left( a + b \sum_{m=1}^{D} \mathbf{q}_{i,m}\mathbf{k}_{N,m} \right) \mathbf{v}_{n,j}, \quad \mathbf{g}_i = \sum_{n=1}^{N} \left( a + b \sum_{m=1}^{D} \mathbf{q}_{i,m}\mathbf{k}_{N,m} \right). \tag{10}$$

Changing the summation orders we get

$$\mathbf{f}_{i,j} = a \sum_{n=1}^{N} \mathbf{v}_{n,j} + b \sum_{m=1}^{D} \sum_{n=1}^{N} \mathbf{q}_{i,m}\mathbf{k}_{N,m}\mathbf{v}_{n,j}, \quad \mathbf{g}_i \quad = a \sum_{n=1}^{N} 1 + b \sum_{m=1}^{D} \sum_{n=1}^{N} \mathbf{q}_{i,m}\mathbf{k}_{N,m}. \tag{11}$$

Applying Factorization we get

$$\mathbf{f}_{i,j} = x_j^{(1)} + \sum_{m=1}^{D} \mathbf{q}_{i,m} x_{jm}^{(2)}, \quad \mathbf{g}_i = y^{(1)} + \sum_{m=1}^{D} \mathbf{q}_{i,m} y_m^{(2)}, \tag{12}$$

where,

$$x_j^{(1)} = a \sum_{n=1}^{N} \mathbf{v}_{n,j}, \quad x_{jm}^{(2)} = b \sum_{n=1}^{N} \mathbf{k}_{N,m}\mathbf{v}_{n,j}, \quad y^{(1)} = a\,N, \quad y_m^{(2)} = b \sum_{n=1}^{N} \mathbf{k}_{N,m}. \tag{13}$$

We have thus demonstrated how to calculate the forward pass of Linear Attention with a computational complexity of $O(ND^2)$. However, when using a differentiable programming library such as JAX or PyTorch, all operations must be tracked in the computational graph. Consequently, all intermediate variables in Eq. 13 need to be stored, resulting in a memory consumption of $O(ND^2)$. See Appendix B on details for how the same memory reduction to the case of a causal mask.

## 3.3 BACKWARD PASS

By deriving the analytical gradient for linear attention, we can calculate the backward pass manually, eliminating the need for differentiable programming libraries. As a result, we no longer need to store intermediate values. However, the backward pass must also be calculated within this same complexity of $O(ND^2)$. Let us denote $\mathbf{q}_i.\mathbf{k}_j$ as $s_{i,j}$, and write the attention and output as

$$\mathbf{o}_{i,j} = \sum_{n=1}^{N} \mathbf{a}_{i,n}\mathbf{v}_{n,j}, \quad \mathbf{a}_{i,n} = \frac{f(s_{i,j})}{\sum_{m=1}^{N} f(s_{i,m})} = \frac{f(s_{i,j})}{\mathbf{g}_i}, \quad f(x) = a + bx \tag{14}$$

Taking the derivative with respect to $s_{i,l}$, we find $\dfrac{\partial \mathbf{o}_{i,j}}{\partial s_{i,l}}$ as

$$\frac{\partial \mathbf{a}_{i,n}}{\partial s_{i,l}} = \begin{cases} \dfrac{b\left(1 - \mathbf{a}_{i,j}\right)}{\sum_{m=1}^{N} f(s_{i,m})}, n = l \\ \dfrac{b\left(-\mathbf{a}_{i,j}\right)}{\sum_{m=1}^{N} f(s_{i,m})}, n \neq l \end{cases} \tag{15}$$

$$\frac{\partial \mathbf{o}_{i,j}}{\partial s_{i,l}} = \sum_{n=1}^{N} \frac{\partial \mathbf{a}_{i,n}}{\partial s_{i,l}} \mathbf{v}_{n,j} = \frac{b\left(\mathbf{v}_{l,j} - \sum_{n=1}^{N} \mathbf{a}_{i,n}\mathbf{v}_{n,j}\right)}{\sum_{m=1}^{N} f(s_{i,m})} = \frac{b}{\mathbf{g}_i}\left(\mathbf{v}_{l,j} - \mathbf{o}_{i,j}\right). \tag{16}$$

We now derive the partial derivative with respect to $\mathbf{Q}, \mathbf{K}, \mathbf{V}$

$$\frac{\partial \mathbf{o}_{i,j}}{\partial \mathbf{q}_{i,r}} = \sum_{l=1}^{N} \frac{\partial s_{i,l}}{\partial \mathbf{q}_{i,r}} \frac{\partial \mathbf{o}_{i,j}}{\partial s_{i,l}} = \frac{\sum_{l=1}^{N} b\,\mathbf{k}_{l,r}}{\mathbf{g}_i}\left(\mathbf{v}_{l,j} - \mathbf{o}_{i,j}\right) \tag{17}$$

$$\frac{\partial \mathbf{o}_{i,j}}{\partial \mathbf{k}_{p,r}} = \frac{\partial s_{i,p}}{\partial \mathbf{k}_{p,r}} \frac{\partial \mathbf{o}_{i,j}}{\partial s_{i,p}} = \frac{b\,\mathbf{q}_{i,r}}{\mathbf{g}_i}\left(\mathbf{v}_{p,j} - \mathbf{o}_{i,j}\right) \tag{18}$$

$$\frac{\partial \mathbf{o}_{i,j}}{\partial \mathbf{v}_{p,j}} = \mathbf{a}_{i,p} = \frac{f(s_{i,p})}{\mathbf{g}_i}. \tag{19}$$

During the backward pass, given the gradient of the previous layer $\mathbf{\Omega}$, the gradient of the Attention head $\nabla \mathbf{\Psi}$ is calculated as follows

$$\nabla_{\mathbf{q}_{i,r}} \mathbf{\Psi} = \sum_{j=1}^{D} \frac{\partial \mathbf{o}_{i,j}}{\partial \mathbf{q}_{i,r}} \mathbf{\Omega}_{i,j} = \sum_{j=1}^{D} \frac{\sum_{l=1}^{N} b\,\mathbf{k}_{l,r}}{\mathbf{g}_i}\left(\mathbf{v}_{l,j} - \mathbf{o}_{i,j}\right)\mathbf{\Omega}_{i,j} \tag{20}$$

$$\nabla_{\mathbf{k}_{p,r}} \mathbf{\Psi} = \sum_{i=1}^{N}\sum_{j=1}^{D} \frac{\partial \mathbf{o}_{i,j}}{\partial \mathbf{k}_{p,r}} \mathbf{\Omega}_{i,j} = \sum_{i=1}^{N}\sum_{j=1}^{D} \frac{b\,\mathbf{q}_{i,r}}{\mathbf{g}_i}\left(\mathbf{v}_{p,j} - \mathbf{o}_{i,j}\right)\mathbf{\Omega}_{i,j} \tag{21}$$

$$\nabla_{\mathbf{v}_{p,j}} \mathbf{\Psi} = \sum_{i=1}^{N} \frac{\partial \mathbf{o}_{i,j}}{\partial \mathbf{v}_{p,j}} \mathbf{\Omega}_{i,j} = \sum_{i=1}^{N} \frac{f(s_{i,p})}{\mathbf{g}_i} \mathbf{\Omega}_{i,j} \tag{22}$$

Applying Factorization, the gradients can be calculated as

$$\nabla_{\mathbf{q}_{i,r}} \mathbf{\Psi} = \sum_{j=1}^{D}\{\alpha_{rj}^Q - \beta_r^Q\,\mathbf{o}_{i,j}\}\mathbf{\Omega}_{i,j}, \quad \nabla_{\mathbf{k}_{i,r}} \mathbf{\Psi} = \sum_{j=1}^{D}\{\alpha_{rj}^K\,\mathbf{v}_{i,j} - \beta_{rj}^K\}, \tag{23}$$

$$\nabla_{\mathbf{v}_{i,j}} \mathbf{\Psi} = \alpha_j^V + \sum_{j=1}^{D}\{\beta_{rj}^V\,\mathbf{k}_{i,r}\}. \tag{24}$$

where the $\alpha$ and $\beta$ are the factorized coefficients and are defined as

$$\alpha_{rj}^Q = \sum_{l=1}^{N} b\,\mathbf{k}_{l,r}\,\mathbf{v}_{l,j}, \quad \beta_r^Q = \sum_{l=1}^{N} b\,\mathbf{k}_{l,r}, \tag{25}$$

$$\alpha_{rj}^K = \sum_{l=1}^{N} b\,\mathbf{q}_{l,r}\,\mathbf{\Omega}_{l,j}, \quad \beta_{rj}^K = \sum_{l=1}^{N} b\,\mathbf{q}_{l,r}\,\mathbf{o}_{l,j}\,\mathbf{\Omega}_{l,j}, \tag{26}$$

$$\alpha_j^V = \sum_{l=1}^{N} a\,\mathbf{\Omega}_{l,j}, \quad \beta_{rj}^V = \sum_{l=1}^{N} b,\,\mathbf{q}_{l,r}\,\mathbf{\Omega}_{l,j}. \tag{27}$$

Put into words, gradient of the output matrix $\mathbf{O}$ can be calculated by storing $\mathbf{Q}, \mathbf{K}, \mathbf{V}, \mathbf{O}$, and $\mathbf{G}$; resulting in a memory consumption of $O(ND)$ elements. The time complexity is $O(ND^2)$ since we perform $O(D)$ operations for $1 \leq i \leq N$, $1 \leq D \leq D$ in Eq.s 68-70, which is the same time complexity of forward pass. To implement custom gradients, we wrote the forward and backward pass using CUDA. However, since the computations are MVP, and that MVP has a low memory reuse rate, the calculations are memory bound. Therefore, we had to implement optimization tricks for improved performance. Appendix C explains these tricks.

## 4 ALTERNATIVE MECHANISM TO DROPOUT

Dropout helps with training transformers by randomly dropping out cells of the Attention matrix during training, which prevents overfitting by reducing the co-adaptation of token pairs, thus improving generalization performance on unseen data. However, since the Attention matrix is not explicitly created in the approaches with linear time complexity, dropout cannot be applied. We propose the following mechanism as an alternative to achieve the same end. Let us denote $\mathbf{O}$ and $\tilde{\mathbf{O}}$ as the output with and without dropout applied. Then, we approximate $\mathbf{O}$ as

$$\mathbf{o}_{ij} \approx (1-p)\tilde{\mathbf{o}}_{ij}(1+\psi), \quad \psi \sim \mathcal{N}(0,p), \quad p : \text{dropout rate.} \tag{28}$$

During the training, we start with some dropout rate $p$, and gradually decrease it to $0$. The intuition behind this mechanism stems from applying the law of large numbers and the central limit theorem when $\mathbf{Q}, \mathbf{K}$, and $\mathbf{V}$ follow a standard Gaussian distribution. For the Softmax-based Attention, dropout is defined as replacing cells of the $\mathbf{Q}\mathbf{K}^T$ Attention matrix with $-\infty$ with probability $p$, and then taking the Softmax to derive the Attention matrix. The Attention matrix is then multiplied by $\mathbf{V}$ to calculate $\mathbf{O}$. Specifically, $\mathbf{O}$ can be written as

$$\mathbf{o}_{ij} = \frac{\sum_{n=1}^{N} \exp(\mathbf{q}_i.\mathbf{k_n}/\sqrt{D})\mathbf{v_{nj}} - \sum_{n\in\{\text{drop}\}} \exp(\mathbf{q}_i.\mathbf{k_n}/\sqrt{D})\mathbf{v_{nj}}}{\sum_{n=1}^{N} \exp(\mathbf{q}_i.\mathbf{k_n}/\sqrt{D}) - \sum_{n\in\{\text{drop}\}} \exp(\mathbf{q}_i.\mathbf{k_n}/\sqrt{D})} = \frac{\eta_{ij} - \eta_{ij}^{\text{drop}}}{\gamma_i - \gamma_i^{\text{drop}}}; \tag{29}$$

$$\eta_{ij} = \sum_{n=1}^{N} \exp(\mathbf{q}_i.\mathbf{k_n}/\sqrt{D})\mathbf{v_{nj}}, \quad \eta_{ij}^{drop} = \sum_{n\in\{\text{drop}\}} \exp(\mathbf{q}_i.\mathbf{k_n}/\sqrt{D})\mathbf{v_{nj}}, \tag{30}$$

$$\gamma_i = \sum_{n=1}^{N} \exp(\mathbf{q}_i.\mathbf{k_n}/\sqrt{D}), \quad \gamma_i^{drop} = \sum_{n\in\{\text{drop}\}} \exp(\mathbf{q}_i.\mathbf{k_n}/\sqrt{D}). \tag{31}$$

As an experiment, let us assume that $\mathbf{Q}, \mathbf{K}$ and $\mathbf{V}$ follow a standard Gaussian distribution, that is

$$\mathbf{q}_{ij} \sim \mathcal{N}(0,1), \quad \mathbf{k}_{ij} \sim \mathcal{N}(0,1), \quad \mathbf{v}_{ij} \sim \mathcal{N}(0,1). \tag{32}$$

**Lemma 4.1.** *(Multiplication of Gaussian Variables) Let $X$ and $Y$ be two independent random variables following standard Gaussian distribution, i.e., $X \sim \mathcal{N}(0,1)$, $Y \sim \mathcal{N}(0,1)$. Then, $Z = XY$ follows a the difference of two Chi-Square distribution with degree of freedom 1.*

*Proof.*

$$XY = \frac{1}{4}((X+Y)^2 - (X-Y)^2); \quad \frac{X+Y}{\sqrt{2}}, \frac{X-Y}{\sqrt{2}} \sim \mathcal{N}(0,1); \tag{33}$$

$$Z = \frac{1}{4}((X+Y)^2 - (X-Y)^2) = \psi - \phi; \quad \psi, \phi \sim \chi_1^2. \tag{34}$$

Furthermore, using the results in Smith et al. (2011), we can write

$$\mathbb{E}[Z] = 0, \quad \mathbb{E}[Z^2] = \frac{1}{2}, \quad Z \sim \Gamma(0, \frac{1}{2}), \tag{35}$$

where $\Gamma(\mu, \sigma^2)$ is some symmetric distribution with mean $\mu$ and variance of $\sigma^2$. $\qquad \square$

Using Lemma 4.1, we write $\mathbf{q}_i.\mathbf{k}_n$ as

$$\mathbf{q}_i.\mathbf{k}_n = \sum_{j=1}^{D} \mathbf{q}_{ij}\mathbf{k}_{nj} = \sum_{j=1}^{D} \omega_j, \quad \omega_j \sim \Gamma(0, \frac{1}{2}). \tag{36}$$

Since $\omega_i$ are i.i.d and follows a symmetric distribution with a bounded variance, we can apply the Central Limit Theorem for sufficiently large $D$, that is

$$\frac{\mathbf{q}_i.\mathbf{k}_n}{\sqrt{D}} = \frac{1}{\sqrt{D}} \sum_{j=1}^{D} \omega_j \sim \mathcal{N}(0, \frac{1}{2}). \tag{37}$$

In practice $D$ is 32 for small transformers, and can reach up to 512. Taking the exponential, we end up with the log-Normal distribution with mean $\exp(\frac{1}{4})$ and variance of 1, that is

$$\exp\left(\frac{\mathbf{q}_i.\mathbf{k}_n}{\sqrt{D}}\right) \sim \text{Log-Normal}(0, \frac{1}{2}). \tag{38}$$

**Lemma 4.2.** *(Kolmogorov's $L_1$ Strong Law of Large Numbers) Let $X_i|_{i=1}^{N}$ be a sequence of i.i.d. random variables with a finite expected value $\mu$. Then,*

$$\lim_{N \to \infty} \sum_{i=1}^{N} X_i \to N\mu \quad \text{almost surely.} \tag{39}$$

In practice, the number of tokens ranges from 512 for small benchmarks, and can reach up to $10^7$. Therefore, we can use Lemma 4.2, to approximate $\gamma_i, \gamma_i^{drop}$ in Eq. 31 as

$$\gamma_i = \sum_{n=1}^{N} \exp\left(\frac{\mathbf{q}_i.\mathbf{k}_n}{\sqrt{D}}\right) \approx N \exp\left(\frac{1}{4}\right), \quad \gamma_i^{drop} = \sum_{n \in \{\text{drop}\}} \exp\left(\frac{\mathbf{q}_i.\mathbf{k}_n}{\sqrt{D}}\right) \approx Np \exp\left(\frac{1}{4}\right), \tag{40}$$

where $p$ is the dropout rate. Since $\mathbf{V} \sim \mathcal{N}(0, I)$, and that sum of Gaussian variables follows a Gaussian distribution, we can use Lemma 4.2 to approximate $\eta_{ij}, \eta_{ij}^{drop}$ in Eq. 31 as

$$\psi_{inj} \sim \mathcal{N}\left(0, \exp\left(\frac{2\mathbf{q}_i.\mathbf{k}_n}{\sqrt{D}}\right)\right) \tag{41}$$

$$\eta_{ij} = \sum_{n=1}^{N} \exp\left(\frac{\mathbf{q}_i.\mathbf{k}_n}{\sqrt{D}}\right)\mathbf{v_{nj}} = \sum_{n=1}^{N} \psi_{inj} \sim \mathcal{N}\left(0, \sum_{n=1}^{N} \exp\left(\frac{2\mathbf{q}_i.\mathbf{k}_n}{\sqrt{D}}\right)\right) \approx \mathcal{N}(0, N \exp(1)) \tag{42}$$

$$\eta_{ij}^{drop} = \sum_{n \in \{\text{drop}\}} \exp\left(\frac{\mathbf{q}_i.\mathbf{k}_n}{\sqrt{D}}\right)\mathbf{v_{nj}} = \sum_{n \in \{\text{drop}\}} \psi_{inj} \sim \mathcal{N}\left(0, \sum_{n \in \{\text{drop}\}} \exp\left(\frac{2\mathbf{q}_i.\mathbf{k}_n}{\sqrt{D}}\right)\right). \tag{43}$$

With these results acquired, we will now write the output with dropout $\mathbf{o}_{ij}$ and without dropout $\tilde{\mathbf{o}}_{ij}$ as

$$\mathbf{o}_{ij} = \frac{\eta_{ij} - \eta_{ij}^{\text{drop}}}{\gamma_i - \gamma_i^{\text{drop}}}, \quad \tilde{\mathbf{o}}_{ij} = \frac{\eta_{ij}}{\gamma_i}. \tag{44}$$

Given $\tilde{\mathbf{o}}_{ij}$, we write $\mathbf{o}_{ij}$ as

$$\mathbf{o}_{ij} = \frac{\gamma_i \, \tilde{\mathbf{o}}_{ij}}{\gamma_i - \gamma_i^{drop}} \frac{\eta_{ij} - \eta_{ij}^{\text{drop}}}{\eta_{ij}} = \frac{\gamma_i \, \tilde{\mathbf{o}}_{ij}}{\gamma_i - \gamma_i^{drop}}\left(1 - \frac{\eta_{ij}^{\text{drop}}}{\eta_{ij}}\right) \approx (1-p)\tilde{\mathbf{o}}_{ij}(1 + \psi); \tag{45}$$

$$\psi \sim \mathcal{N}(0, p). \tag{46}$$

This mechanism can be viewed as adding noise to improve learning by preventing the model from getting stuck in local minima, achieving a result similar to dropout. The concept of adding noise to aid optimization is not new and has been widely employed in various areas. For instance, noise is introduced in environment simulations for reinforcement learning Pinto et al. (2017); Peng et al. (2018), in diffusion probabilistic models Sohl-Dickstein et al. (2015), and in deep networks in general, where adding noise during gradient descent has been suggested to be beneficial Neelakantan et al. (2015); Hu & Gerber (2024).

## 5 EXPERIMENTS

The main contributions of the paper are reducing the memory consumption of Linear Attention and an alternative mechanism to dropout. The expressibility of Linear Attention has been studied in previous works Zhang et al. (2024a); Qin et al. (2022a); Wang et al.; Katharopoulos et al. (2020); Kasai et al. (2021); Qin et al. (2022b); Tay et al.. To confirm the scaling of time and memory of our method, we perform the following experiments: i) We measure the time and memory for a single forward pass of an attention layer in Section 5.1, and ii) for training an LLM in Section 5.2; iii) To assess our alternative mechanism to dropout, we perform an ablation study using two benchmarks in Section 5.3. Our results are for the attention kernel of $f(x) = a + bx$ with $a, b = 1$, but our implementation supports any choice of $a, b$. We use an Nvidia RTX A6000 with 48 GB of memory for all of our experiments, and the implementation details can be found in Appendix D.

## 5.1 TIME AND MEMORY SCALING

We refer to our implementation of Linear Attention as "Ours", and the Kernel Separation approach as "Kern. Sep.". Figure 1 shows the time and memory for a single forward pass for batch size of 32, $H = 16$, $D = 32$, and $1e^3 \leq N \leq 3e^5$. The results are averaged over 1000 runs, and we've included FlashAttention-2 Dao (2023) as a point of comparison with the quadratic conventional algorithm. The plots are in log-log scale, and the slopes show the order of dependency. Both the Kernel Separation and our approach have time and memory scaling of $O(N)$, whereas FlashAttention-2 scales with $O(N^2)$. However, it can be seen that our implementation has $D\times$ less memory consumption compared to the Kernel Separation.

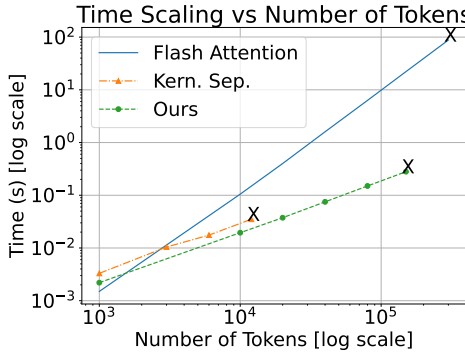 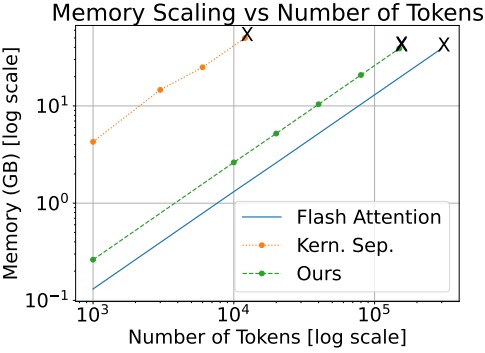

**Figure 1:** Time and memory scaling on A6000 based on number of tokens. The "X" indicate OOM.

Figure 2 shows the time and memory under the same setting but with $N = 2000$ and $32 \leq D \leq 5000$. Both the Kernel Separation and our approach have a time scaling of $O(D^2)$, and FlashAttention-2 has a scaling of $O(D)$. As for the memory, Kernel Separation scales with $O(D^2)$, whereas our approach and FlashAttention-2 scales with $O(D)$. In conclusion, our implementation has a time and memory

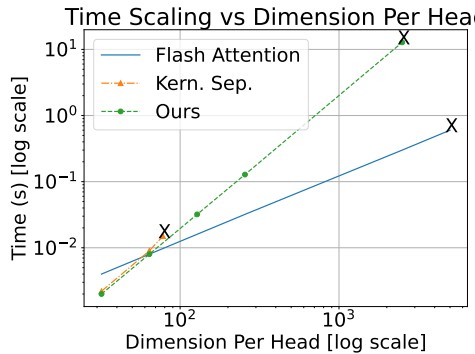 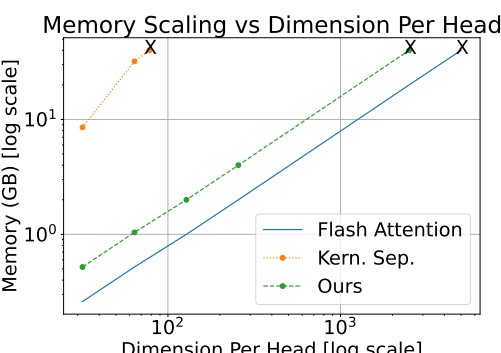

**Figure 2:** Time and memory scaling on A6000 based on dimension per head. The "X" indicate OOM.

scaling linear with $N$ and $D$, whereas Kernel Separation has a memory scaling quadratic with $D$. Our implementation can practically scale to larger contexts.

## 5.2 TRAINING AN LLM

To demonstrate that our claims hold in practice, we train an LLM on a small dataset. We chose Pythia-14M Biderman et al. (2023) as our LLM, with a token length of $N = 4096$, and Tiny Stories Eldan & Li (2023) as our dataset. We train the model using our implementation of Linear Attention, and FlashAttention-2. The hyperparameters used are the same for our method and FlashAttention-2, and more implementation details can be found in Appendix D. Figure 3 shows the learning curves for training loss. Our method has faster convergence, which can be attributed to its linear time scaling of $O(N)$, compared to FlashAttention-2's quadratic scaling of $O(N^2)$. In conclusion, by reducing the

memory cost, our method makes the use of Linear Attention in LLMs and large transformers feasible while maintaining the $O(N)$ time scaling.

## 5.3 DROPOUT ABLATION STUDY

To test the effectiveness of our alternative dropout mechanism introduced in Section 4, we perform an ablation study using the Linear Attention on the Long Range Arena (LRA) Tay et al. and Tiny Stories Eldan & Li (2023) datasets. The LRA benchmark is a suite of tasks designed to evaluate the performance of Transformer models on sequences of extreme length. It includes a diverse set of challenges such as text classification, document retrieval, and mathematical reasoning. The Tiny Stories benchmark is a dataset designed to evaluate language models' ability to generate coherent, child-friendly short stories with simple vocabulary and structure. As shown in Table 1, adding noise to mimic dropout has helped to improve expresivity of Lienar Attention in all cases. We've included the results for Softmax-based attention (FlashAttention-2) with 0.2 dropout as a point of reference.

|  |  | Softmax | Linear | Linear (alt. drop.) |
|---|---|---|---|---|
| LRA | ListOps | 38.37 | 28.55 | 42.76 |
|  | Text | 61.95 | 61.55 | 63.25 |
|  | Retrieval | 80.69 | 65.38 | 78.21 |
|  | Image | 40.57 | 38.47 | 42.76 |
|  | Pathfinder | 65.26 | 50.04 | 66.67 |
| Tiny Stories |  | 3.59 | 3.74 | 3.66 |

**Figure 3 & Table 1:** Figure 3 (left) shows the learning curve of training the Pythia-14M LLM using our method and FlashAttetion-2. Table 1 (right) presents an ablation study comparing our proposed mechanism alternative to dropout for Linear Attention, with Softmax-based attention results included as a benchmark. Our mechanism improves the score in all cases. The score in LRA is accuracy%, and cross-entropy in Tiny Stories.

## 6 CONCLUSION

Transformers use a Softmax-based mechanism known as Attention, which has a quadratic scaling with number of tokens taken in context. Many approaches have proposed subquadratic mechanisms for both training and inference. This problem is particularly important since the current applications are moving towards higher number of tokens, and training can only be done in large datacenter scale systems. The search for a linearly scaling attention mechanism has become an important and active area of research. One of the promising approaches is Linear Attention (Kernel Separation). However, current implementations had very high memory consumption of $O(ND^2)$, where $N$ is the number of tokens and $D$ the model dimension per head. This memory consumption makes kernel separation impractical since even small transformers would require a high amount of memory per batch. We present a solution to reduce the memory scaling of Linear Attention to $O(ND)$. In summary, we derive the analytical format of the attention layer gradient and calculate it in linear time. By implementing this approach instead of relying on a differentiable programming library, we eliminate the need to store the variables for the backward pass, and bring down the memory consumption to $O(ND)$. We provide the details of how to implement the forward and backward pass using Factorization, and confirm the time and memory scaling of our implementation in context of an attention layer and training an LLM. Furthermore, since Linear Attention does not directly compute the attention matrix, dropout cannot be applied. Therefore, we have also introduced an alternative mechanism to dropout, to mimic its effect, and confirmed our mechanism's effectiveness through an ablation study.

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

## A  POLYNOMIAL ATTENTION KERNEL

In Section 3 we showed how to calculate the forward and backward pass with $O(ND^2)$ computations and $O(ND)$ memory for the attention kernel of $f(x) = a + bx$ (Linear Attention). In here we show how to calculate the forward and backward pass for the attention kernel of $f(x) = a + bx + cx^2$ with $O(ND^3)$ computations and $O(ND)$ memory.

### A.1  FORWARD PASS

Employing $f(x) = a + bx + cx^2$ as the attention kernel, the output matrix can be broken down to matrix multiplications in the form of Eq. 4. Specifically,

$$\mathbf{o}_{i,j} = \frac{\sum_{n=1}^{N} f(\mathbf{q}_i^T \mathbf{k_n}) \mathbf{v}_{n,j}}{\sum_{n=1}^{N} f(\mathbf{q}_i^T \mathbf{k_n})}, \quad \mathbf{o}_{i,j} = \frac{\mathbf{f}_{i,j}}{\mathbf{g}_i}, \quad \mathbf{F} \in \mathbb{R}^{N \times D}, \mathbf{G} \in \mathbb{R}^{N}, \tag{48}$$

where $\mathbf{f}_{i,j}$ and $\mathbf{g}_i$ are elements of $\mathbf{F}$ and $\mathbf{G}$. Specifically, $\mathbf{F}$ and $\mathbf{G}$ are written as

$$\mathbf{F} = \left( a + b \sum_{m=1}^{D} \begin{bmatrix} \mathbf{q}_{1,m}\mathbf{k}_{1,m} \dots \mathbf{q}_{1,m}\mathbf{k}_{N,m} \\ \vdots \ddots \vdots \\ \mathbf{q}_{N,m}\mathbf{k}_{1,m} \dots \mathbf{q}_{N,m}\mathbf{k}_{N,m} \end{bmatrix} + \right.$$
$$\left. c \sum_{m,l=1}^{D} \begin{bmatrix} \mathbf{q}_{1,m}\mathbf{k}_{1,m}\mathbf{q}_{1,l}\mathbf{k}_{1,l} \dots \mathbf{q}_{1,m}\mathbf{k}_{N,m}\mathbf{q}_{1,l}\mathbf{k}_{N,l} \\ \vdots \ddots \vdots \\ \mathbf{q}_{N,m}\mathbf{k}_{1,m}\mathbf{q}_{N,l}\mathbf{k}_{1,l} \dots \mathbf{q}_{N,i}\mathbf{k}_{N,i}\mathbf{q}_{N,j}\mathbf{k}_{N,j} \end{bmatrix} \right) \mathbf{V} \tag{49}$$

$$\mathbf{G} = \left( a + b \sum_{m=1}^{D} \begin{bmatrix} \mathbf{q}_{1,m}\mathbf{k}_{1,m} \dots \mathbf{q}_{1,m}\mathbf{k}_{N,m} \\ \vdots \ddots \vdots \\ \mathbf{q}_{N,m}\mathbf{k}_{1,m} \dots \mathbf{q}_{N,m}\mathbf{k}_{N,m} \end{bmatrix} + \right.$$
$$\left. c \sum_{m,l=1}^{D} \begin{bmatrix} \mathbf{q}_{1,m}\mathbf{k}_{1,m}\mathbf{q}_{1,l}\mathbf{k}_{1,l} \dots \mathbf{q}_{1,m}\mathbf{k}_{N,m}\mathbf{q}_{1,l}\mathbf{k}_{N,l} \\ \vdots \ddots \vdots \\ \mathbf{q}_{N,m}\mathbf{k}_{1,m}\mathbf{q}_{N,l}\mathbf{k}_{1,l} \dots \mathbf{q}_{N,i}\mathbf{k}_{N,i}\mathbf{q}_{N,j}\mathbf{k}_{N,j} \end{bmatrix} \right) \mathbb{1} \tag{50}$$

where $\mathbb{1} \in \mathbb{R}^N$ is a vector of all ones; i.e.,

$$\mathbf{f}_{i,j} = \sum_{n=1}^{N} \left( a + b \sum_{m=1}^{D} \mathbf{q}_{i,m}\mathbf{k}_{N,m} + c \sum_{m,l=1}^{D} \mathbf{q}_{i,m}\mathbf{k}_{N,m}\mathbf{q}_{i,l}\mathbf{k}_{N,l} \right) \mathbf{v}_{n,j}, \tag{51}$$

$$\mathbf{g}_i = \sum_{n=1}^{N} \left( a + b \sum_{m=1}^{D} \mathbf{q}_{i,m}\mathbf{k}_{N,m} + c \sum_{m,l=1}^{D} \mathbf{q}_{i,m}\mathbf{k}_{N,m}\mathbf{q}_{i,l}\mathbf{k}_{N,l} \right). \tag{52}$$

Changing the summation orders we get

$$\mathbf{f}_{i,j} = a \sum_{n=1}^{N} \mathbf{v}_{n,j} + b \sum_{m=1}^{D} \sum_{n=1}^{N} \mathbf{q}_{i,m}\mathbf{k}_{N,m}\mathbf{v}_{n,j} + c \sum_{m,l=1}^{D} \sum_{n=1}^{N} \mathbf{q}_{i,m}\mathbf{k}_{N,m}\mathbf{q}_{i,l}\mathbf{k}_{N,l}\mathbf{v}_{n,j}, \tag{53}$$

$$\mathbf{g}_i = a \sum_{n=1}^{N} 1 + b \sum_{m=1}^{D} \sum_{n=1}^{N} \mathbf{q}_{i,m}\mathbf{k}_{N,m} + c \sum_{m,l=1}^{D} \sum_{n=1}^{N} \mathbf{q}_{i,m}\mathbf{k}_{N,m}\mathbf{q}_{i,l}\mathbf{k}_{N,l}. \tag{54}$$

Applying Factorization we get

$$\mathbf{f}_{i,j} = x_j^{(1)} + \sum_{m=1}^{D} \mathbf{q}_{i,m} x_{jm}^{(2)} + \sum_{m,l=1}^{D} \mathbf{q}_{i,m}\mathbf{q}_{i,l} x_{jml}^{(3)}, \quad \mathbf{g}_i = y^{(1)} + \sum_{m=1}^{D} \mathbf{q}_{i,m} y_m^{(2)} + \sum_{m,l=1}^{D} \mathbf{q}_{i,m}\mathbf{q}_{i,l} y_{ml}^{(3)}, \tag{55}$$

where,

$$x_j^{(1)} = a \sum_{n=1}^{N} \mathbf{v}_{n,j}, \quad x_{jm}^{(2)} = b \sum_{n=1}^{N} \mathbf{k}_{N,m} \mathbf{v}_{n,j}, x_{jml}^{(3)} = c \sum_{n=1}^{N} \mathbf{k}_{N,m} \mathbf{k}_{N,l} \mathbf{v}_{n,j}, \tag{56}$$

$$y^{(1)} = a\, N, \quad y_m^{(2)} = b \sum_{n=1}^{N} \mathbf{k}_{N,m}, \quad y_{ml}^{(3)} = v \sum_{n=1}^{N} \mathbf{k}_{N,m} \mathbf{k}_{N,l}. \tag{57}$$

See Appendix B for details on how to apply causal mask.

## A.2 BACKWARD PASS

Let us denote $\mathbf{q}_i.\mathbf{k}_j$ as $s_{i,j}$, and write the attention and output as

$$\mathbf{o}_{i,j} = \sum_{n=1}^{N} \mathbf{a}_{i,n} \mathbf{v}_{n,j}, \quad \mathbf{a}_{i,n} = \frac{f(s_{i,j})}{\sum_{m=1}^{N} f(s_{i,m})} = \frac{f(s_{i,j})}{\mathbf{g}_i}, \quad f(x) = a + bx + cx^2 \tag{58}$$

Taking the derivative with respect to $s_{i,l}$, we find $\dfrac{\partial \mathbf{o}_{i,j}}{\partial s_{i,l}}$ as

$$\frac{\partial \mathbf{a}_{i,n}}{\partial s_{i,l}} = \begin{cases} \dfrac{b + 2c\, s_{i,l}}{\sum_{m=1}^{N} f(s_{i,m})} (1 - \mathbf{a}_{i,j}), n = l \\[3mm] \dfrac{b + 2c\, s_{i,l}}{\sum_{m=1}^{N} f(s_{i,m})} (-\mathbf{a}_{i,j}), n \neq l \end{cases} \tag{59}$$

$$\frac{\partial \mathbf{o}_{i,j}}{\partial s_{i,l}} = \sum_{n=1}^{N} \frac{\partial \mathbf{a}_{i,n}}{\partial s_{i,l}} \mathbf{v}_{n,j} = \frac{b + 2c\, s_{i,l}}{\sum_{m=1}^{N} f(s_{i,m})} \left( \mathbf{v}_{l,j} - \sum_{n=1}^{N} \mathbf{a}_{i,n} \mathbf{v}_{n,j} \right) = \frac{b + 2c\, s_{i,l}}{\mathbf{g}_i} (\mathbf{v}_{l,j} - \mathbf{o}_{i,j}). \tag{60}$$

We now derive the partial derivative with respect to $\mathbf{Q}, \mathbf{K}, \mathbf{V}$

$$\frac{\partial \mathbf{o}_{i,j}}{\partial \mathbf{q}_{i,r}} = \sum_{l=1}^{N} \frac{\partial s_{i,l}}{\partial \mathbf{q}_{i,r}} \frac{\partial \mathbf{o}_{i,j}}{\partial s_{i,l}} = \frac{\sum_{l=1}^{N} (b + 2c\, s_{i,l}) \mathbf{k}_{l,r}}{\mathbf{g}_i} (\mathbf{v}_{l,j} - \mathbf{o}_{i,j}) \tag{61}$$

$$\frac{\partial \mathbf{o}_{i,j}}{\partial \mathbf{k}_{p,r}} = \frac{\partial s_{i,p}}{\partial \mathbf{k}_{p,r}} \frac{\partial \mathbf{o}_{i,j}}{\partial s_{i,p}} = \frac{(b + 2c\, s_{i,p}) \mathbf{q}_{i,r}}{\mathbf{g}_i} (\mathbf{v}_{p,j} - \mathbf{o}_{i,j}) \tag{62}$$

$$\frac{\partial \mathbf{o}_{i,j}}{\partial \mathbf{v}_{p,j}} = \mathbf{a}_{i,p} = \frac{f(s_{i,p})}{\mathbf{g}_i}. \tag{63}$$

$$\tag{64}$$

During the backward pass, given the gradient of the previous layer $\mathbf{\Omega}$, the gradient of the Attention head $\nabla \mathbf{\Psi}$ is calculated as follows

$$\nabla_{\mathbf{q}_{i,r}} \mathbf{\Psi} = \sum_{j=1}^{D} \frac{\partial \mathbf{o}_{i,j}}{\partial \mathbf{q}_{i,r}} \mathbf{\Omega}_{i,j} = \sum_{j=1}^{D} \frac{\sum_{l=1}^{N} (b + 2c\, s_{i,l}) \mathbf{k}_{l,r}}{\mathbf{g}_i} (\mathbf{v}_{l,j} - \mathbf{o}_{i,j}) \mathbf{\Omega}_{i,j} \tag{65}$$

$$\nabla_{\mathbf{k}_{p,r}} \mathbf{\Psi} = \sum_{i=1}^{N} \sum_{j=1}^{D} \frac{\partial \mathbf{o}_{i,j}}{\partial \mathbf{k}_{p,r}} \mathbf{\Omega}_{i,j} = \sum_{i=1}^{N} \sum_{j=1}^{D} \frac{(b + 2c\, s_{i,p}) \mathbf{q}_{i,r}}{\mathbf{g}_i} (\mathbf{v}_{p,j} - \mathbf{o}_{i,j}) \mathbf{\Omega}_{i,j} \tag{66}$$

$$\nabla_{\mathbf{v}_{p,j}} \mathbf{\Psi} = \sum_{i=1}^{N} \frac{\partial \mathbf{o}_{i,j}}{\partial \mathbf{v}_{p,j}} \mathbf{\Omega}_{i,j} = \sum_{i=1}^{N} \frac{f(s_{i,p})}{\mathbf{g}_i} \mathbf{\Omega}_{i,j} \tag{67}$$

Applying Factorization, the gradients will be calculated as

$$\nabla_{\mathbf{q}_{i,r}} \mathbf{\Psi} = \sum_{j=1}^{D} \{\alpha_{rj}^{Q} - \beta_{r}^{Q} \mathbf{o}_{i,j} + \sum_{m=1}^{D} (\gamma_{rjm}^{Q} - \zeta_{rm}^{Q} \mathbf{o}_{i,j}) \mathbf{q}_{i,m}\} \mathbf{\Omega}_{i,j} \tag{68}$$

$$\nabla_{\mathbf{k}_{i,r}} \mathbf{\Psi} = \sum_{j=1}^{D} \{\alpha_{rj}^{K} \mathbf{v}_{i,j} - \beta_{rj}^{K} + \sum_{m=1}^{D} (\gamma_{rmj}^{K} \mathbf{v}_{i,j} - \zeta_{rmj}^{K}) \mathbf{k}_{i,m}\} \tag{69}$$

$$\nabla_{\mathbf{v}_{i,j}} \mathbf{\Psi} = \alpha_{j}^{V} + \sum_{j=1}^{D} \{\beta_{rj}^{V} \mathbf{k}_{i,r} + \sum_{m=1}^{D} \gamma_{rmj}^{V}, \mathbf{k}_{i,r}, \mathbf{k}_{i,m}\}. \tag{70}$$

where the $\alpha$, $\beta$, $\gamma$ and $\zeta$ are the factorized coefficients and are defined as

$$\alpha_{rj}^{Q} = \sum_{l=1}^{N} b\, \mathbf{k}_{l,r}\, \mathbf{v}_{l,j}, \quad \beta_{r}^{Q} = \sum_{l=1}^{N} b\, \mathbf{k}_{l,r}, \tag{71}$$

$$\gamma_{rmj}^{Q} = \sum_{l=1}^{N} 2c\, \mathbf{k}_{l,r} \mathbf{k}_{l,m} \mathbf{v}_{l,j}, \quad \zeta_{rm}^{Q} = \sum_{l=1}^{N} 2c\, \mathbf{k}_{l,r} \mathbf{k}_{l,m} \tag{72}$$

$$\alpha_{rj}^{K} = \sum_{l=1}^{N} b\, \mathbf{q}_{l,r}\, \mathbf{\Omega}_{l,j}, \quad \beta_{rj}^{K} = \sum_{l=1}^{N} b\, \mathbf{q}_{l,r}\, \mathbf{o}_{l,j}\, \mathbf{\Omega}_{l,j}, \tag{73}$$

$$\gamma_{rmj}^{K} = \sum_{l=1}^{N} 2c\, \mathbf{q}_{l,r}\, \mathbf{q}_{l,m}\, \mathbf{\Omega}_{l,j}, \quad \zeta_{rmj}^{K} = \sum_{l=1}^{N} 2c\, \mathbf{q}_{l,r}\, \mathbf{q}_{l,m}\, \mathbf{o}_{l,j}\, \mathbf{\Omega}_{l,j} \tag{74}$$

$$\alpha_{j}^{V} = \sum_{l=1}^{N} a\, \mathbf{\Omega}_{l,j}, \quad \beta_{rj}^{V} = \sum_{l=1}^{N} b,\, \mathbf{q}_{l,r}\, \mathbf{\Omega}_{l,j}, \quad \gamma_{rmj}^{V} = \sum_{l=1}^{N} 2c\, \mathbf{q}_{l,r}\, \mathbf{q}_{l,m}\, \mathbf{\Omega}_{l,j} \tag{75}$$

## B  APPLYING CAUSAL MASK

For the sake of space, we only show the process for applying causal mask for the attention kernel of $f(x) = a + bx + cx^2$. To apply the causal mask for Linear Attention ($f(x) = a + bx$), we merely need to set $c = 0$ in the equations below.

To apply causal mask, we change Eq. 48 to

$$\mathbf{o}_{i,j} = \frac{\sum_{n=1}^{i} f(\mathbf{q}_i^T \mathbf{k_n}) \mathbf{v}_{n,j}}{\sum_{n=1}^{i} f(\mathbf{q}_i^T \mathbf{k_n})}, \quad \mathbf{o}_{i,j} = \frac{\mathbf{f}_{i,j}}{\mathbf{g}_i}, \quad \mathbf{F} \in \mathbb{R}^{N \times D}, \mathbf{G} \in \mathbb{R}^{N}, \tag{76}$$

where $\mathbf{F}$ and $\mathbf{G}$ are

$$\mathbf{f}_{ij} = \sum_{n=1}^{i} (a + b \sum_{m=1}^{D} \mathbf{q}_{im} \mathbf{k}_{nm} + c \sum_{m,l=1}^{D} \mathbf{q}_{im} \mathbf{k}_{nm} \mathbf{q}_{il} \mathbf{k}_{nl}) \mathbf{v}_{nj}, \tag{77}$$

$$\mathbf{g}_{i} = \sum_{n=1}^{i} (a + b \sum_{m=1}^{D} \mathbf{q}_{im} \mathbf{k}_{nm} + c \sum_{m,l=1}^{D} \mathbf{q}_{im} \mathbf{k}_{nm} \mathbf{q}_{il} \mathbf{k}_{nl}), \tag{78}$$

where we changed the first summation range. Changing the summation orders and applying Factorization we get

$$\mathbf{f}_{ij} = x_{j}^{(1)} + \sum_{m=1}^{D} \mathbf{q}_{im} x_{jm}^{(2)} + \sum_{m,l=1}^{D} \mathbf{q}_{im} \mathbf{q}_{il} x_{jml}^{(3)}, \tag{79}$$

$$\mathbf{g}_{i} = y^{(1)} + \sum_{m=1}^{D} \mathbf{q}_{im} y_{m}^{(2)} + \sum_{m,l=1}^{D} \mathbf{q}_{im} \mathbf{q}_{il} y_{ml}^{(3)}, \tag{80}$$

where,

$$x_{1j}^{(1)} = a\, \mathbf{v}_{1j}, \quad x_{ij}^{(1)} = x_{i-1j}^{(1)} + a\, \mathbf{v}_{ij},$$

$$x_{1jm}^{(2)} = b\, \mathbf{k}_{1m}\mathbf{v}_{1j}, \quad x_{ijm}^{(2)} = x_{i-1jm}^{(2)} + b\, \mathbf{k}_{im}\mathbf{v}_{ij},$$

$$x_{1jml}^{(3)} = c\, \mathbf{k}_{1m}\mathbf{k}_{1l}\mathbf{v}_{1j}, \quad x_{ijml}^{(3)} = x_{i-1jml}^{(3)} + c\, \mathbf{k}_{im}\mathbf{k}_{il}\mathbf{v}_{ij}, \tag{81}$$

$$y_i^{(1)} = a\, i, \quad y_{1m}^{(2)} = b\, \mathbf{k}_{1m}, \quad y_{im}^{(2)} = y_{i-1m}^{(2)} + b\, \mathbf{k}_{im},$$

$$y_{1ml}^{(3)} = c\, \mathbf{k}_{1m}\mathbf{k}_{1l}, \quad y_{iml}^{(3)} = y_{i-1ml}^{(3)} + c\, \mathbf{k}_{im}\mathbf{k}_{il}. \tag{82}$$

## C  CUDA Implementation

To implement custom gradients, we wrote the forward and backward pass of the Attention head in CUDA. As a reminder, the matrices $\mathbf{Q}, \mathbf{K}, \mathbf{V}, \mathbf{O}$ are 4-dimensional tensors with dimensions batch, Attention heads, tokens, and dimension per head. We parallelized our code within the batch, heads, and dimension per head, meaning we lunch a total of $B \times H \times D$ threads. For a CUDA code to be efficient, the number of threads should be either an integer multiple of number of CUDA cores (hardware dependant), or be much bigger. For small benchmarks, we have $B \geq 32$, $H \geq 8$, $D \geq 32$ resulting in launching $8,192$ threads. For comparison, the NVIDIA A100 GPU has $6,912$ CUDA cores, meaning this method of parallelization is sufficient for reaching maximum parallelization, even for small benchmarks. Note that it is not efficient for the calculations to be parallelized within the tokens dimensions, since in the case of Attention with casual mask, the factorized coefficients for each token is calculated with respect to the previous token's, as shown in Eq. 8182.

Since the computations are MVP, and that MVP has a low memory reuse rate, the calculations are memory bound. To alleviate this issue, we take advantage of shared memory. Shared memory, is cache-like memory, which means it has a much lower read/write time but has a very limited capacity (in order of 100 KB for each thread-block). As the name suggests, shared memory is shared between the threads within each thread-block. The first trick we implement is to use all of the threads within a thread-block to fill the shared memory, and then each thread can use all of the loaded data. To elaborate, assume we have thread-block of $n$ threads, and $c$ variables are needed by each thread. Assume accessing the global and shared memory takes $t_g$ and $t_s$ seconds for each variable, where $t_g >> t_s$. Without the trick, the time needed would be $c \times t_g$ since each thread accesses global memory $c$ times for the calculations. However, using the trick, the time will be $\frac{c}{n} \times t_g + (c + \frac{c}{n}) \times t_s$ since each thread accesses the global and shared memory $\frac{c}{n}$ times to fill the shared memory, and accesses the shared memory $c$ times for the calculations.

Even by using shared memory, the calculation are memory bound. Therefore, we shape the calculations in a way to take advantage of the available registers within each thread. Registers have a much lower read/write time compared to shared memory memory, but has limited availability (less than 1 KB per thread). Registers are useful when a variable needs to be updated many times within each thread. Instead of accessing this variable from shared memory, the thread stores and updates it locally, and writes it in shared memory when done. To elaborate, assume accessing the shared memory and register takes $t_s$ and $t_r$ seconds for each variable, where $t_s >> t_r$, and that a variable needs to be updated $d$ times. The time without and with using this trick will be $d \times t_s$ and $t_s + d \times t_r$ respectively.

## D  Experiment Implementation Details

### D.1  Details of Section 5.2

We have used the LitGPT AI (2023) framework to implement and train our Pythia-14M LLM Biderman et al. (2023). The training was done using the Tiny Stories dataset Eldan & Li (2023) with FlashAttention-2 **?** (Softmax-based attention) and our method (Linear Attention). The token length is 4096, and the hyperparamters are the same for both of the implementations, and both implementations use RoPE Su et al. (2024). The precision is Float-32, and the learning rate is $1e^{-3}$. The epoch count and batch size is 1 and 25, and dropout rate is 0.2 (see Section 4 for details on how we mimic dropout for Linear Attention).

## D.2 DETAILS OF SECTION 5.3

The Softmax is implemented using FlashAttention-2, and dropout is applied.

| Benchmark | Embedding Dimension | Num Layers | Num Heads | Learning Rate | Dropout Rate |
|---|---|---|---|---|---|
| LRA | 64 | 2 | 2 | $7e^{-4}$ | 0.1 |
| Tiny Stories | 128 | 6 | 4 | $1e^{-3}$ | 0.2 |

**Table 2:** We used the standard LRA model configuration (number of layers, embedding dimension, etc.) as required for each of the LRA tests. For Tiny Stories, we used Pythia-14M.

