# OpenReview forum: "Towards Making Linear Attention Usable"
_ICLR.cc/2025/Conference — Submitted to ICLR 2025_

### Official Review · Reviewer_Umfs · 2024-10-19

**Soundness:** 2
**Presentation:** 2
**Contribution:** 2
**Rating:** 5
**Confidence:** 3

**Summary:**

This paper focus on reduce the space complexity of vanilla linear attention. To be specific, the authors change the order of some matrix multiplication in the forward pass of linear attention when the kernel is linear. The authors also provide a detailed derivation of the forward pass and backward pass. Because linear attention does not have a explicit attention map in the forward pass, the authors propose an alternative for the dropout regularization mechanism. Some testing experiments shows this method are indeed save the memory footprint, but the experiment is not abundant.

**Strengths:**

1. The topic of saving the memory footprint of linear attention is interesting. This is important while it seems few researcher have noticed.

2. The forward and backward derivation is also very clean.

3. An alternative for dropout is proposed, and the experiments in LRA and Tiny Stories datasets verifies its effectiveness.

**Weaknesses:**

(1) After the forward and backward modification, althought the computational seems equivalent compared with linear attention, will the proposed method achieve the same performance is still unclear. The authors can check the RepVGG paper, which shows that equivalent computational may not produce the same performance.

(2) The paper only change is computing order of Linear Attn, and proposed a new dropout alternative. I think the novelty is limited.

(3) Lack of experiment.

    (3.1) for the time and memory scaling experiments (sec. 5.1). The authors only show one fixed number of head H, fixed head dim D, varying token length result experiment, and, one fixed token length N, varying head dim D experiment. I think the author should show more results on the H, D choices, since this experiment is training free, and whether the experiment results would show the same trend. Additionally, the only chosen one is weird. Since in ViT-Base H=12 and D=64, in DiT-S H=6 and D=64, I cannot find a popular model with H=16 and D=32.

    (3.2) For LLM experiment in Sec. 5.2, the author does not express the reason why they choose this model and this dataset. And they only show the loss curve (also the orange curve is not trained as long as the blue one is). No evaluation metric provided.

    (3.3) For the ablation study in Sec. 5.3, I do not get the point why the author do not provide these accuracy results in main results, but put them in ablation. The training hypermeter is also not clear. I have also check the LRA results in other papers (e.g. S4, by Albert Gu in ICLR 2022, https://arxiv.org/pdf/2111.00396). The results of LRA in the submission are far lower than this 2 years ago paper. And both the Softmax result and Linear(alt. drop.) results have different numerical range compared with the results in S4 paper.

(4) The experiment hyper-parameter is not described. That may be the reason why it cannot match the results in S4 paper. I think the authors should follow others setting, or explain why they choose a different setting.

(5)Minors:

    (5.1) wrong formula in line 90: f(x) = exp(q \cdot k / root(D)), while there is no x in the expression.

    (5.2) a latex bug in appendix D.1 (line 860) have not be fixed.

**Questions:**

1. In section 3.1, the final operation we need to calculate is

[a_1, a_2, ..., a_n]^T  \times [b_1, b_2, ..., b_n] \times [d_1, d_2, ..., d_n]^T

If we compute the first \times first, there will be n^2+n^2 multiplications and n*(n-1) accumulations. If we compute the second \times first, there will be (n+n) multiplications and (n-1) accumulateions.

So the point sec 3.1 made it that, change the operation order will save FLOPs. Do I miss something?

2. The  new dropout alternative seems complicated. It seems no experiments discuss its complexity. Will it take much inference latency?

3. Overall, I give my initial rating as "marginally below the acceptance threshold". I hope the authors can solve my concerns as I mentioned in weakness and questions section.

---

> ### Author Response · Authors · 2024-11-20
>
> Dear Reviewer Umfs, we thank you for your comments. We believe that our main contribution or its importance has not been recognized. We humbly ask you to read the below explanation, reconsider your assessment. We have addressed your concerns afterwards.
>
>
> There are at least 20 papers introducing some sort of Linear Attention, each with computational complexity of $O(N)$ and showing an expressivity comparable with softmax-based attention. However, we have yet to see a single Linear Attention mechanism being utilized in a real-life LLM. The reason is the high memory overhead which almost none of them mention. To be specific, implementing a causal mask for Linear Attention is complicated since the attention matrix is not created to maintain the $O(N)$ time. As explained in [3], the linear attention with causal mask requires processing $O(ND^2)$. When using an automatic differentiation library such as Pytorch, all of the operations need to be stored in the computational graph, meaning the memory complexity of Linear Attention with causal mask will be $O(ND^2)$. This means a small model such as TinyLlama would need ~200GB of memory, making Linear Attention impractical. We derive the analytical gradient of attention heads, find how it can be calculated in $O(N)$ time, implement it using thousands of lines of CUDA, and tune the parallelization and data movement so well that it outperforms Pytorch’s implementation. Using our code, we no longer need to store the operations in the computational graph, and we only need to store the Q, K, V and layer output which reduces the memory complexity to $O(ND)$.
>
>
> Our main contribution is introducing a library which enables the implementation of any Linear Attention (an attention mechanism with $ax+b$ kernel) with and without causal mask with $O(ND)$ memory. Similar to how FlashAttention [4] efficiently implements softmax-based attention and reduced the memory cost from $O(N^2)$ to $O(ND)$ using analytical gradients, we implement Linear Attention and reduce the memory cost from $O(ND^2)$ to $O(ND)$. We are paving the path for the use of Linear Attention, which is well within the criteria of ICLR’s Infrastructure category. If the memory cost of Linear Attention remains $O(ND^2)$, its potential cannot be studied, and it can never be implemented for real-life applications.
>
>
> Regarding questions:
> 1. In this example changing the matrix multiplication is the same as factorization. In connection to the Linear Attention, without applying causal mask we could change the multiplication order and the result would be $O(N)$ as well. However, If a causal mask is applied, the calculations can no longer be expressed as a simple matrix multiplication, so reordering matrix multiplications won't work. In Appendix B we show how the calculation pattern for Linear Attention with a causal mask can be factorized.
>
>
> 2. The dropout mustn't be used during inference, it should be used only during training to improve the training. As for the computational complexity, it barely affects the process. To elaborate, to apply our dropout, the code is as follows $o[i][j] *= (1+p*s[i][j])$, where $o$ is the calculated output matrix, $p$ the dropout rate and $s[i][j]$ are IID Normal variables. This results in $O(ND)$ operations. To calculate the output matrix the process takes $O(ND^2)$ calculations.
>
>
> Regarding weaknesses:
>
>
> (1) Our code calculates the same exact value as the pytorch implementation of Linear Attention, and we have checked it for both forward and backward pass using pytorch’s “gradcheck”.
>
>
> (2) Implementing a causal mask for Linear Attention is complicated since the attention matrix is not created to maintain the $O(N)$ time. Several works [1,2] have tried to somewhat mimic a causal mask. Not only do their suggestions introduce computational and memory overheads, but also it’s not the same as causal mask. We derive the analytical gradient of attention heads, find how it can be calculated in $O(N)$ time, implement it using thousands of lines of CUDA, and tune the parallelization and data movement so well that it outperforms Pytorch’s implementation. Using our code, we no longer need to store the operations in the computational graph, and we only need to store the Q, K, V and layer output which reduces the memory complexity to $O(ND)$. How is this not enough contribution?
>
> (continued in the next comment)

---

> > ### Author Response · Authors · 2024-11-20
> >
> > (3.1) That’s a good point, and we will attach more experiments with various D by the rebuttal deadline.
> >
> >
> > (3.2) Expressivity is not the purpose of our paper. The expressivity and quality of Linear Attention has been studied by tens of papers and should not be under consideration here. What should be considered is that we are significantly reducing  the memory cost without increasing time. Yes we could repeat the experiments that have been reported in previous papers and show Linear Attention performs well, but what’s the point? Our goal is to demonstrate how Linear Attention can be implemented efficiently. We have provided analytical proof of our time and complexity claim, and confirmed it by evaluating the time and memory of the attention layer, as well as an end to end setting.
> > (3.3) Our main claim is the implementation of Linear Attention with reduced memory cost, and therefore the main results demonstrate the time and memory. The ablation study’s goal is to demonstrate the effectiveness of our dropout. Regarding the difference between our result and the mentioned paper, we are using the LRA implementation of Nystromformer [5], and we’re using their default model parameters. We also performed a hyperparameter search for the lr.
> >
> >
> > (4) The hyperparameters we used are mentioned in Appendix D.
> >
> > [1] When Linear Attention Meets Autoregressive Decoding: Towards More Effective and Efficient Linearized Large Language Models.
> > [2] Medusa: Simple LLM Inference Acceleration Framework with Multiple Decoding Heads.
> > [3] Transformer Quality in Linear Time.
> > [4] FlashAttention: Fast and Memory-Efficient Exact Attention with IO-Awareness.
> > [5] https://github.com/mlpen/Nystromformer/tree/main

---

> ### Comment · Reviewer_Umfs · 2024-11-23
>
> Appreciate the author's reply.
>
> 1. For weakness 1: Since the gradient is the same, I believe it will produce the same training result as the vanilla linear attention.
>
> 2. For weakness 2: thanks for the authors' contribution. Now I am aware their contribution to the ML community. Do you have any plan to release your CUDA code or integrate your code into pytorch library if your paper is accepted？Do you have future plan to make more types of linear attention usage?
>
> 3. For weakness 3: I am eager to see your result regarding weakness 3.1, since abundant experiment results and analysis are still important in a research paper like ICLR. I think testing the memory cost and inference latency does not need training. It will be very fast to get these results.
>
> 4. For weakness 4: I think it is better to make the training hyper-parameter, the inference latency testing configuration, and other details clear in the revision.
>
> 5. For weakness 5: The authors do not reply to this point. And the pdf file also has no modification.
>
> 6. For question 1: I think it is a writing problem. The authors should make more effort to make it easy to comprehend.
>
> 7. For question 2: For the dropout-like method, I still think a detailed experiment and analysis is required for an ICLR paper.
>
> Overall, I'm positive for this submission. If the authors will provide detailed experiments, analysis, take more effort in writing, and will release their code, I will raise my rating to weak accept.

---

### Official Review · Reviewer_y3jG · 2024-10-22

**Soundness:** 2
**Presentation:** 2
**Contribution:** 2
**Rating:** 3
**Confidence:** 4

**Summary:**

This manuscript proposes two techniques to improve the practical efficiency of linear attention. One is the factorization of matrix multiplication, and the other is a modified dropout technique. Toy experiments show the effectiveness of the two techniques.

**Strengths:**

- The Motivation is clear.

- The theoretical derivation is detailed.

**Weaknesses:**

Overall, this manuscript is not ready for publication. My concerns are listed as follows:

- The technical novelty is limited. The first proposed method, matrix multiplication multiplication, is a standard operation in linear attention methods [1]. I see some novelty in the second dropout technique, but its effectiveness is validated only in small-scale experiments.

- The experiments are not convincing enough. One small model (Pythia-14M) is adopted, and only a training curve is presented. To make the results more convincing, it is recommended to conduct experiments on larger-scale models (both language and vision models should be included) and well-known benchmarks. The comparison should be the about the trade-off between efficiency and performance.

- The presentation needs substantial improvements. Some examples are listed as follows:.
  - In the introduction, the authors categorize linear attention methods into two lines of work:  Sparsified/localized Attention, and Kernel Separation. However, no references are cited here.
  - In the equations (e.g. eq1), the index subscriptions $i,n,j$ should not have bold fonts.
  - In Sec. 3.1, the FLOPs calculation example could be given by normal matrix presentations instead of a 3x3 toy example.
  - The derivaton in Sections 3&4 are too detailed and might distract readers from the main method. It is recommended to put detailed derivation process in the appendix, and present the major method in the main text.


[1] Flatten transformer: Vision transformer using focused linear attention. ICCV, 2023.

**Questions:**

How does the proposed method compare with those RNN-like methods such as Mamba and RetNet?

---

> ### Author Response · Authors · 2024-11-20
>
> Dear Reviewer y3jG, we thank you for your comments. We believe that our main contribution or its importance has not been recognized. We humbly ask you to read the below explanation, reconsider your assessment. We have addressed your concerns afterwards.
>
>
> There are at least 20 papers introducing some sort of Linear Attention, each with computational complexity of $O(N)$ and showing an expressivity comparable with softmax-based attention. However, we have yet to see a single Linear Attention mechanism being utilized in a real-life LLM. The reason is the high memory overhead which almost none of them mention. To be specific, implementing a causal mask for Linear Attention is complicated since the attention matrix is not created to maintain the $O(N)$ time. As explained in [3], the linear attention with causal mask requires processing $O(ND^2)$. When using an automatic differentiation library such as Pytorch, all of the operations need to be stored in the computational graph, meaning the memory complexity of Linear Attention with causal mask will be $O(ND^2)$. This means a small model such as TinyLlama would need ~200GB of memory, making Linear Attention impractical.
> We derive the analytical gradient of attention heads, find how it can be calculated in $O(N)$ time, implement it using thousands of lines of CUDA, and tune the parallelization and data movement so well that it outperforms Pytorch’s implementation. Using our code, we no longer need to store the operations in the computational graph, and we only need to store the Q, K, V and layer output which reduces the memory complexity to $O(ND)$.
>
>
> Our main contribution is introducing a library which enables the implementation of any Linear Attention (an attention mechanism with $ax+b$ kernel) with and without causal mask with $O(ND)$ memory. Similar to how FlashAttention [4] efficiently implements softmax-based attention and reduced the memory cost from $O(N^2)$ to $O(ND)$ using analytical gradients, we implement Linear Attention and reduce the memory cost from $O(ND^2)$ to $O(ND)$. We are paving the path for the use of Linear Attention, which is well within the criteria of ICLR’s Infrastructure category. If the memory cost of Linear Attention remains $O(ND^2)$, its potential cannot be studied, and it can never be implemented for real-life applications.
>
>
> - Regarding limited contribution, implementing a causal mask for Linear Attention is complicated since the attention matrix is not created to maintain the $O(N)$ time. Several works [1,2] have tried to somewhat mimic a causal mask. Not only do their suggestions introduce computational and memory overheads, but also it’s not the same as causal mask. We derive the analytical gradient of attention heads, find how it can be calculated in $O(N)$ time, implement it using thousands of lines of CUDA, and tune the parallelization and data movement so well that it outperforms Pytorch’s implementation. Using our code, we no longer need to store the operations in the computational graph, and we only need to store the Q, K, V and layer output which reduces the memory complexity to $O(ND)$. How is this not enough contribution? As for Flatten transformer, they are not introducing any method to reduce the memory overhead. In fact, they have not mentioned their memory.
>
>
> - Regarding limited evaluation, The expressivity and quality of Linear Attention has been studied by tens of papers and should not be under consideration here. What should be considered is that we are significantly reducing  the memory cost without increasing time. Yes we could repeat the experiments that have been reported in previous papers and show Linear Attention performs well, but what’s the point? Our goal is to demonstrate how Linear Attention can be implemented efficiently. We have provided analytical proof of our time and complexity claim, and confirmed it by evaluating the time and memory of the attention layer, as well as an end to end setting.
>
>
> - We apologize for the hastiness, and thank you for listing some of the issues.
>
>
> [1] When Linear Attention Meets Autoregressive Decoding: Towards More Effective and Efficient Linearized Large Language Models.
> [2] Medusa: Simple LLM Inference Acceleration Framework with Multiple Decoding Heads.
> [3] Transformer Quality in Linear Time.
> [4] FlashAttention: Fast and Memory-Efficient Exact Attention with IO-Awareness.

---

### Official Review · Reviewer_Tg66 · 2024-11-04

**Soundness:** 2
**Presentation:** 2
**Contribution:** 2
**Rating:** 3
**Confidence:** 4

**Summary:**

This paper addresses two major limitations in current linear attention mechanisms. First, although existing approaches aim to reduce computational complexity to linear time, they still practically require O(ND²). To address this, this work proposes a method to lower the memory usage to O(ND). By deriving and computing the attention layer gradient analytically, this paper achieves this efficiency without relying on a differentiable programming library, thus avoiding the need to store variables for backpropagation. This work uses Factorization in both forward and backward passes and validates the reduced memory and time complexity in the context of attention layers and large language model training. Additionally, because linear attention doesn’t inherently compute an attention matrix, dropout cannot be directly applied. To overcome this, this work introduces an alternative mechanism that emulates the effect of dropout, with its effectiveness confirmed through an ablation study.

**Strengths:**

In terms of practicality, reducing memory cost is very crucial. If this work maintains comparable performance or minimizes the performance drop, it has impactful potential.

**Weaknesses:**

### Lacks of benchmarks
Although it shows memory and latency experiments by increasing token length, it didn't show common LLM benchmarks such as MMLU and GLUE or long-context LLM benchmarks such as n streaming books (PG19), Long Context Understanding (LongBench), and book summarization (Booksum).

**Questions:**

Same Weakness

---

> ### Author Response · Authors · 2024-11-20
>
> Dear Reviewer Tg66, we thank you for your comments. We believe that our main contribution or its importance has not been recognized. We humbly ask you to read the below explanation, reconsider your assessment. We have addressed your concerns afterwards.
>
>
> There are at least 20 papers introducing some sort of Linear Attention, each with computational complexity of $O(N)$ and showing an expressivity comparable with softmax-based attention. However, we have yet to see a single Linear Attention mechanism being utilized in a real-life LLM. The reason is the high memory overhead which almost none of them mention. To be specific, implementing a causal mask for Linear Attention is complicated since the attention matrix is not created to maintain the $O(N)$ time. As explained in [3], the linear attention with causal mask requires processing $O(ND^2)$. When using an automatic differentiation library such as Pytorch, all of the operations need to be stored in the computational graph, meaning the memory complexity of Linear Attention with causal mask will be $O(ND^2)$. This means a small model such as TinyLlama would need ~200GB of memory, making Linear Attention impractical. We derive the analytical gradient of attention heads, find how it can be calculated in $O(N)$ time, implement it using thousands of lines of CUDA, and tune the parallelization and data movement so well that it outperforms Pytorch’s implementation. Using our code, we no longer need to store the operations in the computational graph, and we only need to store the Q, K, V and layer output which reduces the memory complexity to $O(ND)$.
>
>
> Our main contribution is introducing a library which enables the implementation of any Linear Attention (an attention mechanism with $ax+b$ kernel) with and without causal mask with $O(ND)$ memory. Similar to how FlashAttention [4] efficiently implements softmax-based attention and reduced the memory cost from $O(N^2)$ to $O(ND)$ using analytical gradients, we implement Linear Attention and reduce the memory cost from $O(ND^2)$ to $O(ND)$. We are paving the path for the use of Linear Attention, which is well within the criteria of ICLR’s Infrastructure category. If the memory cost of Linear Attention remains $O(ND^2)$, its potential cannot be studied, and it can never be implemented for real-life applications.
>
>
> - We have received a reject score since we are not performing benchmarks to evaluate the expressivity of Linear Attention. Expressivity is not the purpose of our paper. The expressivity and quality of Linear Attention has been studied by tens of papers and should not be under consideration here. What should be considered is that we are significantly reducing  the memory cost without increasing time. Yes we could repeat the experiments that have been reported in previous papers and show Linear Attention performs well, but what’s the point? Our goal is to demonstrate how Linear Attention can be implemented efficiently. We have provided analytical proof of our time and complexity claim, and confirmed it by evaluating the time and memory of the attention layer, as well as an end to end setting.

---

### Official Review · Reviewer_gUD9 · 2024-11-12

**Soundness:** 2
**Presentation:** 2
**Contribution:** 1
**Rating:** 3
**Confidence:** 4

**Summary:**

This paper attempts to make linear attention efficient and useable by reducing the memory consumption and introducing an alternative mechanism to dropout. Results show the usefulness of the proposed approaches, while maintaining the linear scaling in N in both wall-clock time and memory usage.

**Strengths:**

- The paper is well written and easy to follow.
- Interesting topic -- making linear attention efficient is of great interest to the research community.

**Weaknesses:**

- The results are somewhat unsatisfactory, with only limited to a very small model. More experiments and analysis are needed to justify the effectiveness of the proposed method.
- What about practical usage? Can this method start from the pre-trained LLMs and directly convert them to efficient linear attention? Also, can it start from a LLM with linear attention and make it efficient through little amount of training or finetuning?
- Can this method scale to large models? Say, billion scale models. Authors at least perform experiments using 1-3B models.
- What about hardware efficiency of the proposed changes to the linear attention?

**Questions:**

I’d like to rate the current submission reject due to limited technical contributions and lack of convincing experiments. The paper needs significant changes including new experiments and possibly methodological improvements in justifying the practical use behind the proposed method.

---

> ### Author Response · Authors · 2024-11-20
>
> Dear Reviewer gUD9, we thank you for your comments. We believe that our main contribution or its importance has not been recognized. We humbly ask you to read the below explanation, reconsider your assessment. We have addressed your concerns afterwards.
>
>
> There are at least 20 papers introducing some sort of Linear Attention, each with computational complexity of $O(N)$ and showing an expressivity comparable with softmax-based attention. However, we have yet to see a single Linear Attention mechanism being utilized in a real-life LLM. The reason is the high memory overhead which almost none of them mention. To be specific, implementing a causal mask for Linear Attention is complicated since the attention matrix is not created to maintain the $O(N)$ time. As explained in [3], the linear attention with causal mask requires processing $O(ND^2)$. When using an automatic differentiation library such as Pytorch, all of the operations need to be stored in the computational graph, meaning the memory complexity of Linear Attention with causal mask will be $O(ND^2)$. This means a small model such as TinyLlama would need ~200GB of memory, making Linear Attention impractical. We derive the analytical gradient of attention heads, find how it can be calculated in $O(N)$ time, implement it using thousands of lines of CUDA, and tune the parallelization and data movement so well that it outperforms Pytorch’s implementation. Using our code, we no longer need to store the operations in the computational graph, and we only need to store the Q, K, V and layer output which reduces the memory complexity to $O(ND)$.
>
>
> Our main contribution is introducing a library which enables the implementation of any Linear Attention (an attention mechanism with $ax+b$ kernel) with and without causal mask with $O(ND)$ memory. Similar to how FlashAttention [4] efficiently implements softmax-based attention and reduced the memory cost from $O(N^2)$ to $O(ND)$ using analytical gradients, we implement Linear Attention and reduce the memory cost from $O(ND^2)$ to $O(ND)$. We are paving the path for the use of Linear Attention, which is well within the criteria of ICLR’s Infrastructure category. If the memory cost of Linear Attention remains $O(ND^2)$, its potential cannot be studied, and it can never be implemented for real-life applications.
>
>
>
>
> - Regarding limited contribution, implementing a causal mask for Linear Attention is complicated since the attention matrix is not created to maintain the $O(N)$ time. Several works [1,2] have tried to somewhat mimic a causal mask. Not only do their suggestions introduce computational and memory overheads, but also it’s not the same as causal mask. We derive the analytical gradient of attention heads, find how it can be calculated in $O(N)$ time, implement it using thousands of lines of CUDA, and tune the parallelization and data movement so well that it outperforms Pytorch’s implementation. Using our code, we no longer need to store the operations in the computational graph, and we only need to store the Q, K, V and layer output which reduces the memory complexity to $O(ND)$. How is this not enough contribution?
>
>
> - Regarding limited evaluation, expressivity is not the purpose of our paper. The expressivity and quality of Linear Attention has been studied by tens of papers and should not be under consideration here. What should be considered is that we are significantly reducing  the memory cost without increasing time. Yes we could repeat the experiments that have been reported in previous papers and show Linear Attention performs well, but what’s the point? Our goal is to demonstrate how Linear Attention can be implemented efficiently. We have provided analytical proof of our time and complexity claim, and confirmed it by evaluating the time and memory of the attention layer, as well as an end to end setting.
>
> - Regarding using a pretrained model, if a model is trained using Linear Attention, then yes, our code can be used as “Plug and Play”. There is no need for finetuning as our code can be used to implement any attention with an attention kernel of ax+b (Linear Attention).
>
>
> - Regarding Larger models, all of our claims are proven analytically. The model size wouldn’t make a difference. For the implementational aspect, our code has $D$ threads per block, $B\times H\times D$ blocks, uses 1KB of shared memory per block, and each thread performs $O(ND)$ operations. The only limitation is for the model to fit within the GPU (our approach reduces the memory from O(ND^2) to O(ND)). To be specific, our memory consumption for a model with $K$ parameters, $L$ layers, $H$ heads, $D$ dimension per head, and $N$ tokens is $12K + 60LHDN$ Bytes. For context, a model with LLama3.2-1B parameters and token length of 4096 can fit within a single A6000.
>
>
> (continued in the next comment)

---

> > ### Author Response · Authors · 2024-11-20
> >
> > - Regarding hardware efficiency, it varies based on the model parameters and context length, and does not measure the throughput of the algorithm. For example a larger number of heads will improve the parallelization, and therefore will result in a higher hardware efficiency.
> >
> >
> >
> >
> > [1] When Linear Attention Meets Autoregressive Decoding: Towards More Effective and Efficient Linearized Large Language Models.
> > [2] Medusa: Simple LLM Inference Acceleration Framework with Multiple Decoding Heads.
> > [3] Transformer Quality in Linear Time.
> > [4] FlashAttention: Fast and Memory-Efficient Exact Attention with IO-Awareness.

---

### Meta-Review · Area_Chair_DE5C · 2024-12-08

**Metareview:**

The paper proposes methods to enhance the efficiency of linear attention by reducing memory complexity and introducing an alternative dropout mechanism.

Strengths:
* Relevance: The focus on reducing memory complexity in linear attention aligns well with the infrastructure challenges in modern LLMs.
* Technical Clarity: The derivations of memory and time complexities are detailed and well-structured.
* Potential Contribution: Enabling efficient linear attention with causal masking could unlock broader applications in real-world scenarios.

Weaknesses:
* Lack of Comprehensive Evaluation: The paper relies heavily on small-scale experiments (e.g., Pythia-14M) and does not include evaluations on larger models or widely accepted benchmarks (e.g., MMLU, LongBench).
* Expressivity and performance trade-offs, particularly for large models, remain unaddressed, making the practical utility of the proposed method unclear.
* Limited Novelty: The reordering of matrix multiplications is presented as a key contribution but is considered a standard operation in the linear attention literature. The proposed dropout alternative is inadequately validated, with minimal discussion of its computational overhead or performance benefits in broader contexts.
* Insufficient Experimental Validation: The ablation studies and scaling analyses are limited in scope and fail to provide convincing evidence for the method's efficacy across different model sizes and configurations. The experimental setup lacks clear alignment with prior work, leading to inconsistencies in reported results.
* Presentation Issues: Writing and presentation suffer from organizational and typographical errors, detracting from readability and clarity.
Detailed derivations dominate the main text, which could be better presented in appendices to improve accessibility.

Overall, while the proposed memory optimizations could be impactful, the current submission lacks the experimental rigor, novelty, and practical validation required for acceptance at ICLR. Future work should address these deficiencies by incorporating large-scale evaluations, aligning with standard benchmarks, and refining the presentation.

**Additional Comments On Reviewer Discussion:**

Points Raised by Reviewers:

gUD9:
Limited experiments with small models, lacking practical application and scalability to larger models.
Questions about converting pre-trained LLMs to linear attention and hardware efficiency.

Tg66:
Absence of common LLM benchmarks to validate performance in practical scenarios.

y3jG:
Limited novelty, with techniques like factorization seen as standard.
Insufficient experiments, particularly with larger models and vision tasks.

Umfs:
Concerns about the equivalence of performance despite computational equivalence.
Limited experimental validation, discrepancies in LRA results compared to prior work.
Presentation issues, including equation formatting and clarity.

Authors' Responses:

General Response:
Emphasized memory reduction as their primary contribution, comparing it to FlashAttention's impact on softmax attention.
Clarified that their aim wasn't to measure expressivity but to enable practical use of linear attention.

Specific Points:

gUD9:
Explained implementation details for large models and memory reduction, but did not provide new experiments for larger models.

Tg66:
Argued that benchmarks for expressivity were not the focus, instead providing analytical proof for memory and time complexity.

y3jG:
Asserted novelty in memory reduction, not in factorization, and defended against claims of limited contribution.

Umfs:
Confirmed computational equivalence and planned to add more experiments on different model sizes but did not address all presentation issues.

---

### Decision · Program_Chairs · 2025-01-22

Reject